# Indoor Robot Path Planning Using an Improved Whale Optimization Algorithm

**DOI:** 10.3390/s23083988

**Published:** 2023-04-14

**Authors:** Qing Si, Changyong Li

**Affiliations:** School of Mechanical Engineering, Xinjiang University, Urumqi 830039, China

**Keywords:** indoor robotics, path planning, whale optimization algorithm, nonlinear convergence factor, Corsi variance

## Abstract

An improved whale optimization algorithm is proposed to solve the problems of the original algorithm in indoor robot path planning, which has slow convergence speed, poor path finding ability, low efficiency, and is easily prone to falling into the local shortest path problem. First, an improved logistic chaotic mapping is applied to enrich the initial population of whales and improve the global search capability of the algorithm. Second, a nonlinear convergence factor is introduced, and the equilibrium parameter A is changed to balance the global and local search capabilities of the algorithm and improve the search efficiency. Finally, the fused Corsi variance and weighting strategy perturbs the location of the whales to improve the path quality. The improved logical whale optimization algorithm (ILWOA) is compared with the WOA and four other improved whale optimization algorithms through eight test functions and three raster map environments for experiments. The results show that ILWOA has better convergence and merit-seeking ability in the test function. In the path planning experiments, the results are better than other algorithms when comparing three evaluation criteria, which verifies that the path quality, merit-seeking ability, and robustness of ILWOA in path planning are improved.

## 1. Introduction

Path planning is an important part of the indoor robotics research process, and the quality of the path is one of the most important factors for the robot to be able to move efficiently and autonomously to perform related tasks, which is directly related to the robot’s mobility efficiency. As the scope of robot applications continues to expand, in recent years from industrial environments to logistics and warehousing, home services, catering industry and other indoor environment applications, the requirements for robot navigation and movement continue to rise. Many intelligent algorithms have been applied to the field of mobile navigation for robots, such as common biological heuristics: ant colony algorithm [1,2], particle swarm algorithm [3], gray wolf algorithm [4], etc. Compared with traditional non-heuristic algorithms, intelligent optimization algorithms rely less on mathematical models, are easy to understand, and have better global optimization finding ability. It is widely used in indoor robot path planning and has very important research significance.

The traditional whale optimization algorithm (WOA) is widely favored since it has the advantages of fewer parameters, a simple structure, and is easy to understand and apply. In recent years, it is applied in model prediction [5,6], fault diagnosis [7,8], optimization design [9,10], path planning [11,12], scheduling problems [13,14], etc. However, the algorithm also has some defects, such as poor population initialization ability, low population diversity, a poor balance between global search ability and local search ability, low search efficiency and accuracy, easily falls into a local optimum, etc. Qiang Zhang [15] used the local search ability and global search ability of the inertial weight balancing algorithm to discretize whale individuals by an improved Sigmoid function to increase the richness of the population, but at the same time aggravate the computational power of the algorithm and increase the computation time. Wu Kun [16] introduced the hierarchy in the whale optimization algorithm and the greedy strategy in the differential evolution algorithm in solving the UAV path planning problem based on the whale algorithm to improve the exploitation and search capability of the algorithm. Jingnan LI [17] introduced an adaptive nonlinear inertia weight based on the Branin function to balance global and local search, and proposed a mirror selection method to improve population richness and convergence speed. Dingli Chu [18] proposed an adaptive weighting strategy and introduced the simulated annealing algorithm into the whale algorithm, which improved the convergence speed and global merit-seeking ability of the algorithm. Wenqiang Yang [19] proposed an exploratory prey mechanism and designed a constraint processing strategy to improve the population diversity of the algorithm and increase the global search capability. Kumar S [20] combined fuzzy logic techniques with the WOA algorithm to design a hybrid path planning algorithm and applied it to the static and dynamic path planning of robots. Simulation and real machine experiments in MATLAB verified the effectiveness of the improvement and improved the distance by about 20.63% compared to other algorithms that have been improved. Yaonan Dai [21] proposed a new whale optimization algorithm (NWOA), which designs virtual obstacles and introduces adaptive techniques to solve the problems of slow convergence and easy to fall into local optimization in robot path planning by the original algorithm and other two improved algorithms for experiments, and the results showed that the path planning time and the average lengths of the path of the NWOA are as short as possible. Yucen Cai [22] used the secondary optimization of the harmonic search algorithm to improve the quality and global search ability of the population and improve the search accuracy, and introduced a dynamic balancing strategy and a population reconstruction mechanism to regulate the global search ability and local search ability of the algorithm to avoid getting into local optimal solutions. The improved algorithm shows obvious advantages in path optimality, stability, and convergence speed in path planning when conducting path planning experiments in different environments. Weijun Zhang [23] proposed a discrete whale optimization algorithm (DWOA) for indoor logistics robot path planning and established an AGV path planning model, and through simulation experiments, the transportation time of AGV was shortened and the transportation efficiency was improved.

For the above analysis of the improvement of the whale optimization algorithm, this paper proposes a multi-strategy fusion improved whale optimization algorithm. An improved logistic chaotic mapping is used to initialize the whale population strategy to enrich the diversity of the population. A sinusoidal weighting strategy is introduced with the adjustment balance parameter A to enhance the adaptive ability of the algorithm in global search and local search. A nonlinear factor is introduced in the bracketing mechanism of the algorithm and a cosine weighting strategy is introduced in the spiral mechanism to perturb the whale position to avoid falling into the local optimum in the local area and to accelerate the convergence speed and convergence accuracy of the algorithm. Finally, in MATLAB software, comparative experiments were conducted using eight test functions and comparative experiments on path planning simulations using three indoor environments of varying complexity to verify the effectiveness of the improved application of the whale optimization algorithm to path planning and improve the path quality of the robot when moving indoors.

The rest of this paper is structured as follows: Section 2 provides a detailed introduction of the whale algorithm; Section 3 provides a brief analysis of the shortcomings of the whale algorithm as well as a proposed improvement strategy; Section 4 conducts test function experiments on the improved whale algorithm and five other improved whale algorithms, and analyzes the mean, standard deviation, and iteration count results of the solutions, proving that the improved whale algorithm has better performance in finding the best solution. Indoor raster maps of three different complex environments are built and path planning simulation experiments are conducted to verify the effectiveness and feasibility of the improved whale algorithm in solving indoor path planning problems. Finally, Section 5 summarizes the conclusions and identifies the next steps.

## 2. The Whale Optimization Algorithm

The whale optimization algorithm (WOA) [24] is a novel population intelligence optimization algorithm proposed by Mirjalili scholars in 2016, which is inspired by the foraging behavior of humpback whales and is divided into three main types: (1) swimming and foraging, (2) surround foraging, and (3) attacking prey. Wandering foraging and encircling predation are determined by the regulation coefficient A, where A takes values in the range [−2, 2]. A schematic diagram of the whale optimization algorithm is shown in Figure 1.

When 0 ≤ |*A*| ≤ 1, the algorithm performs a wandering encirclement mechanism or a surround predation mechanism; when 1 ≤ |*A*| ≤ 2, the algorithm performs a prey search mechanism.

(1) Swim away encirclement. Assuming that the optimal solution in the current generation population is the target prey and all individual whales swim toward the optimal position, the mathematical model is as follows:(1)$$X(t+1)=X^{*}(t)-A\times |C\times X^{*}(t)-X(t)|$$
where X(t) denotes the position of individual whales in the first-generation population, $X^{*}(t)$ denotes the optimal position of the whale individuals in the first-generation population, and A,C denote the adjustment coefficient. The specific expressions are as follows:(2)$$A=2a\times r_{1}-a$$
(3)$$C=2\times r_{2}$$
where a denotes the convergence factor that decreases linearly from 2 to 0, and $r_{1},r_{2}$ denotes a random number between [0, 1]. The specific expression is as follows:(4)$$a=2-2\times \frac{t}{t_{\max}}$$
where t denotes the number of current iterations. $t_{max}$ denotes the maximum number of iterations.

(2) Surround predation. After spotting prey, humpback whales spiral up in preparation for an attack, with the following position update expression:(5)$$X(t+1)=X^{*}(t)+D_{p}\times e^{b\times l}\times \cos(2\times \pi \times l)$$
where l denotes a random number between [−1, 1], b denotes the constant coefficient defining the shape of the logarithmic spiral, and $D_{p}$ denotes the distance between a whale in the current generation and the current optimal individual, as expressed by the following expression:(6)$$D_{p}=|X^{*}(t)-X(t)|$$

As the whale spirals around the prey, it must also swim away to surround the prey. To express this process, Mirjalili scholars assumed that the probability p of a humpback whale choosing both swim-and-surround and surround predation is 50%, which is represented by the mathematical model:(7)$$X(t+1)=\left\{ \begin{array}{l} \begin{array}{cc} X^{*}(t)-A\times |C\times X^{*}(t)-X(t)| & p<0.5 \end{array}\\ \begin{array}{cc} X^{*}(t)+D_{p}\times e^{b\times l}\times \cos(2\times \pi \times l) & p\geq 0.5 \end{array} \end{array}\right.$$

(3) Searching for prey. When 0 ≤ |*A*| ≤ 1, whales move by probabilistically choosing to swim away to surround and encircle their prey. When 1 ≤ |*A*| ≤ 2, instead of approaching the current generation of optimal individuals as prey, the humpback whale selects a random whale individual in the population as prey for roundup, and the specific mathematical model expression is as follows:(8)$$X(t+1)=X_{rand}(t)-A\times |X_{rand}(t)-X(t)|$$
where $X_{rand}(t)$ indicates the location of a random individual whale in the t generation population.

## 3. The Problem Statement and Improvement Were Measured

Initializing population strategies based on chaotic mapping. Initialization of whale population strategy using improved logistic chaos mapping. The original whale optimization algorithm uses a random distribution of whale populations, resulting in uneven distribution and poor diversity of the initial populations, leading to slow convergence of the algorithm and low convergence accuracy, reducing the search performance of the algorithm and leading to lower speed and poorer accuracy of robot path planning. Chaotic mapping with good disorder and ergodicity is used to initialize the whale population, enrich the diversity of the population, and improve the full specific exploration ability of the algorithm in a certain range. Logistic chaos mapping, which is often used in intelligent bionic algorithms to initialize populations, applies logistic chaos mapping to the algorithm with the following specific mathematical model expression:(9)X(t+1)=r×X(t)×(1−X(t))
where X(t) denotes the location of individual whales in the t generation population, *t* denotes the number of iterations, *r* denotes the random number of [0, 4].

Figure 2 and Figure 3 show the distribution and histogram of the chaotic mapping values of the original logistic in the interval [0, 1], respectively. The mapping values are relatively more distributed near the two endpoints in the interval [0, 1], the distribution in the middle is sparse and uneven, the initial population distribution applied in the algorithm is relatively uneven, and the population diversity is not significantly enhanced.

Therefore, an improved logistic chaotic mapping is proposed, as shown in Figure 4 and Figure 5. The improved logistic chaotic mapping values are more uniformly distributed, the percentage of each region in the interval [0, 1] does not differ much from each other, and the initialized particles are more evenly distributed. The chaotic sequence generated by the improved logistic chaos mapping is introduced into the search space of whale individuals to generate the sequence of individual positions in the initial stage of the whale optimization algorithm to improve the initial population distribution of whales, whose mathematical model expression is
(10)$$X(t+1)=\left\{ \begin{array}{l} \lambda_{1}\times X(t)\times (1-X(t))+\lambda_{1}\times (4-\lambda_{1})\times X(t)/2\begin{array}{cccc} & if & t<0.5 & \end{array}\\ \lambda_{2}\times X(t)\times (1-X(t))+\lambda_{2}\times (4-\lambda_{2})\times (1-X(t))/2\begin{array}{ccc} & if & t\geq 0.5 \end{array} \end{array}\right.$$
where X(t) denotes the location of individual whales in the tth generation of the population, t denotes the number of iterations. X ∈ [0, 1]. $\lambda_{1},\lambda_{2}$ indicates the adjustment coefficient in the range of [0, 4]; all values are 0.3.

Next, the balancing strategy using nonlinear convergence factors is discussed. In the original whale optimization algorithm, the balancing parameter *A* is used to regulate the local search capability and the global search capability of the algorithm. The original algorithm has weak global search capability, and to increase the global exploration capability of the algorithm, the share of global exploration is expanded. After several tests adjusting the *A*-value to 1.3 (a 15% improvement in the percentage), the algorithm has a stronger global exploration capability. The value of *A* varies depending on the linear factor. Since the value of a decreases linearly from 2 to 0, it leads to slow convergence of the algorithm, weak global exploration ability, and low convergence accuracy. Therefore, a nonlinear convergence strategy is introduced to reduce the speed of decreasing the value of the improved a at the beginning of the iteration compared with the value of the original a. This ensures that larger values are used at the beginning of the iteration to increase the global exploration capability of the algorithm and improve the search accuracy; the value decreases at a faster rate at the end of the iteration and the local search capability is enhanced to speed up the convergence of the algorithm, as shown in Figure 6.The specific mathematical model expression for the improved convergence factor is as follows:(11)$$a=a_{max}-(a_{max}-a_{min})\times \sin((\frac{1}{2}\times (t/t_{max}{)}^{2})\times \pi )$$
where $a_{max}$ indicates the maximum value within the range of values taken for *a*, which is 2; $a_{min}$ indicates the smallest value within the range of values taken for *a*, which is 0.

The nonlinear factorial balancing strategy, to a certain extent, improves the global search performance of the algorithm, and because the value of A is affected by the random numbers in Equations (2) and (3), the local search capability of the algorithm is affected, and the line accuracy of the robot path planning is reduced. Therefore, a weighting strategy is introduced in the position update formula for global and local search of the nonlinear factors in collaboration to improve the local search capability of the algorithm, with the following specific weight expressions:(12)$$\phi (t)=1-{\left(\frac{t}{t_{\max}}\right)}^{\lambda }$$
where *λ* indicates the adjustment coefficient; here the value is taken as 3. The improved position update equations are, respectively,
(13)$$X(t+1)=\phi (t)\times X^{*}(t)-A\times |C\times X^{*}(t)-X(t)|$$
(14)$$X(t+1)=\phi (t)\times X^{*}(t)+D_{p}\times e^{b\times l}\times \cos(2\times \pi \times l)$$
(15)$$X(t+1)=\phi (t)\times X_{rand}(t)-A\times |X_{rand}(t)-X(t)|$$

Next, the location update strategy is adjusted by using the Corsi variant. When the WOA is updated with the position after the above strategy improvement, it relies on the new position after each iteration without active perturbation update of the target position, and it is easy to fall into local optimum in the late iteration, which leads the robot to fall into local shortest path in path planning. Therefore, based on this, the Corsi variation strategy is introduced to perform a random perturbation update on the target position, which helps to improve the algorithm’s search capability and search accuracy and avoid falling into local optimum. The improved position update expression is as follows:(16)$$X(t+1)=X(t)+{\left[Cauchy\times A\right]}^{\alpha }$$
where α denotes the variance factor, which here takes the value of 1. Cauchy is the standard Cauchy distribution, and the specific variable generating function is
(17)Cauchy=tan((rand(1,dim)−0.5)×π)

The specific workflow of ILWOA is shown in Figure 7.

## 4. Verification and Analysis of Simulation Experiments

In order to test the improvement effect of ILWOA in robot path planning, the improved algorithm is verified in the standard test function and path planning, respectively. Experimental simulation platform: Windows 10 computer with model 3.0 GHz, 8 GB RAM, Intel(R) Core(TM) i5-8500U CPU, and software MATLAB R2018a.The algorithm parameters in the simulation experiments are set as shown in Table 1. To verify the objectivity and accuracy of the path planning comparison experiments, the population size is set uniformly as N, the maximum number of iterations as $t_{max}$, the search dimension as dim, the nonlinear factors $a_{max},a_{min}$, and the balance parameter *A* in the ILWOA algorithm.

### 4.1. Standard Test Function Experiment

To detect the improved performance of ILWOA algorithm, the ILWOA algorithm is compared with traditional WOA, MWOA, TWOA, IWOA, and AWOA algorithms in standard test function comparison experiments. The specific parameter settings of the five algorithms are shown in Table 1, and they are tested by eight standard test functions in single-peak, multi-peak, and fixed dimensions, and the specific test functions are shown in Table 2, and the convergence curves are shown in Figure 8. Thirty experiments were run in eight test functions and the standard deviation and average of the experimental results were taken, as shown in Table 3.

As shown in Figure 8, the horizontal coordinate represents the number of iterations of the algorithm, and the vertical coordinate represents the optimal fitness value; the lower fitness value represents the better ability of the algorithm to solve the problem and the more superior performance. Among the eight test functions, the red line represents the 500 iterations of the ILWOA algorithm, and from Figure 8a–e,g, it can be concluded that the ILWOA algorithm can escape from the local optimum after 500 iterations, the accuracy of the search for the optimum is improved, the value of the fitness of the solution is the lowest, and the ability of the search for the optimum is the strongest. From Figure 8f,h, it can be seen that all algorithms find the same fitness value after 500 iterations, but the ILWOA finds the optimal value after less than 20 iterations, and the algorithm has an improved search efficiency.

As shown in Table 3, the mean and standard deviation were obtained from 30 experiments of six improved whale correlation algorithms in eight test functions. By comparing the data, the average fitness values and standard deviations solved 30 times by the ILWOA algorithm are larger compared with the results of other improved algorithms and are the lowest values, verifying that the stability of the improved algorithms is significantly improved.

### 4.2. Path Planning

The feasibility and effectiveness of the improvement of the ILWOA algorithm is initially verified through the experiments of the test function. The algorithm is applied to path planning, which is also the process of solving. In order to further verify the feasibility of the ILWOA algorithm for solving path planning problems, this section uses raster maps as a simulation environment for robot path planning to verify the experiments comparing the ILWOA with other algorithms on improved applied path planning. First, the function solved is replaced by the algorithm with a raster map, the population of whales is redistributed in the two-dimensional raster map using Formula (10), the length of whales from the target point is calculated and the whales closest to the target point are determined. Then, the whales start to search the target location of the raster map, according to the change in the A value. The whales search according to Formulas (13)–(15) in three ways; Equation (16) is updated for the position accordingly, and the nearest whale to the target point is determined again, and the cycle is reiterated until a whale finds the target point and all whales approach it. When the number of iterations is equal to 500, the whales stop searching, and finally, the shortest path from the initial point to the target point is found according to the paths taken by all whales. Three indoor environments with different levels of complexity were designed, as shown in Figure 9, Figure 10, Figure 11, Figure 12 and Figure 13, for the specific simulated indoor maps of Environment 1 (a small indoor area with a small level of environmental complexity in a 15 × 15 grid), Environment 2 (a medium indoor area with a medium level of environmental complexity in a 25 × 25 grid), and Environment 3 (a large indoor area with a large level of environmental complexity in a 40 × 40 grid), all with the upper left corner set as the starting point and the lower right corner set as the end point. The average path length, standard deviation, and average number of iterations are used as metrics to evaluate the quality of path planning.

As shown in Figure 9 and Figure 10, in the 15 × 15 grid environment, the number of obstacles set on the optimal path is small and the shape is simple, and the optimal results of 30 path planning experiments are the same, but the ILWOA algorithm finds the optimal path after five iterations, with fewer iterations, and the planning efficiency of ILWOA in robot path planning is improved. This is due to the introduction of an adaptive weighting strategy and nonlinear factor, balancing the weight of local search and global search, accelerating the convergence speed, improving the search efficiency, and enhancing the global search capability.

As shown in Figure 11 and Figure 12, in a 25 × 25 raster environment with a medium number of obstacle settings and complex shapes on the optimal path, among the optimal results of 30 path planning experiments, the shortest path length found by the ILWOA algorithm is smaller than the other five algorithms, and the number of path turns is smaller, and the path quality is improved. This is attributed to the introduction of the Corsi variation strategy, which makes the algorithm jump out of the local optimum by random position perturbation and improves the accuracy of the algorithm.

As shown in Figure 13 and Figure 14, in the 40 × 40 grid environment, the number of obstacles set on the optimal path is more dense and the search area is larger. In the optimal results of 30 path planning experiments, the number of iterations and the distance when the shortest path is found by the ILWOA algorithm are smaller than the other five algorithms, which verifies that the robot’s ability to find the optimal in a complex indoor environment is better than the other authors’ improvements. This is made possible by applying an improved logistic chaos mapping and resetting the value of the equilibrium parameter A, which enriches the diversity of the population, increases the global search capability, search efficiency and robustness of the algorithm, and improves the robot’s ability to fully explore the planned path.

As shown in Table 4, Table 5 and Table 6, after 30 independent experiments based on three different environments, the average path length of the ILWOA algorithm for the raster map ILWOA algorithm at 15 × 15 is 30.667, the average number of iterations is 36 and the standard deviation of the path is 0.956.The average path length of the ILWOA algorithm for the raster map at 25 × 25 is 53.467, with an average number of 73 iterations and a path standard deviation of 1.479.The average path length of the raster map ILWOA algorithm at 40 × 40 is 90.667, the average number of iterations is 98, and the standard deviation of the path is 2.746. By comparing the average path length, path standard deviation, and average number of iterations, the ILWOA application shows some advantages in robot path planning. It can be seen that ILWOA has a better comprehensive merit-seeking ability in path planning problems compared to those of other authors.

## 5. Conclusions and Future Work

In this study, an improved whale optimization algorithm (ILWOA) is proposed to solve the global path planning problem for indoor robots. To address the shortcomings of the uneven population distribution of the WOA algorithm, an improved logistic chaotic mapping is introduced to initialize the population, improve the global search capability of the algorithm, and increase the path finding capability of the robot. To address the problem of incoordination between global exploration and local exploration of the WOA algorithm, a nonlinear convergence factor is introduced and the value of the equilibrium parameter A is increased to improve the comprehensive exploration capability, speed up the convergence speed and accuracy, and enhance the accuracy and efficiency of robot path planning. Since the WOA algorithm easily falls into the local optimum, the position update weight strategy and the Corsi variation strategy are used to randomly jump the updated position to avoid falling into the local optimum and enhance the robot to find the best path in the entire planning path environment. The standard test function and indoor global path planning simulation experiments verify the following:

(1) In the comparison to the standard test function, the ILWOA has a significant advantage in the number of iterations, convergence speed, and optimization-seeking accuracy, verifying the effectiveness of the improvement.

(2) In the path planning comparison experiments under three different environments, the ILWOA’s ability to find optimal paths, stability, environmental adaptability, and convergence efficiency in path planning are improved more than other comparison algorithms. After the above two verifications, ILWOA has certain superiority in indoor path planning in terms of seeking ability, stability of seeking results, convergence speed, and path quality, and solves the problems of a poor path, a long planning time, and the bottom efficiency of the WOA algorithm in robot path planning.

For future work, we will next investigate the smoothing of the ILWOA’s routes in robot path planning, as well as combining the algorithm with other algorithms to further demonstrate the research and application value of the algorithm.

## Figures and Tables

**Figure 1 sensors-23-03988-f001:**
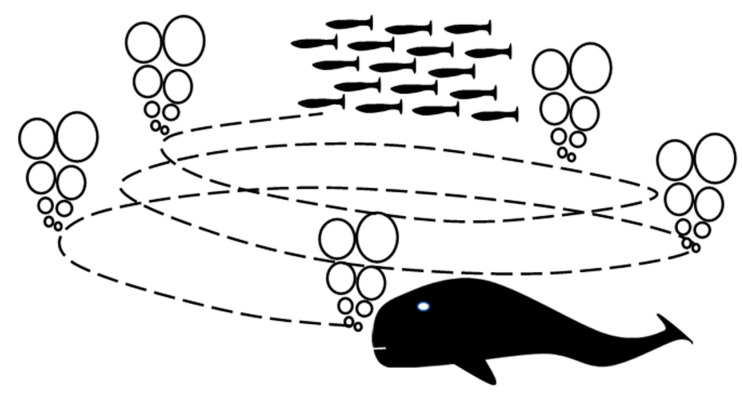
Schematic diagram of the whale optimization algorithm.

**Figure 2 sensors-23-03988-f002:**
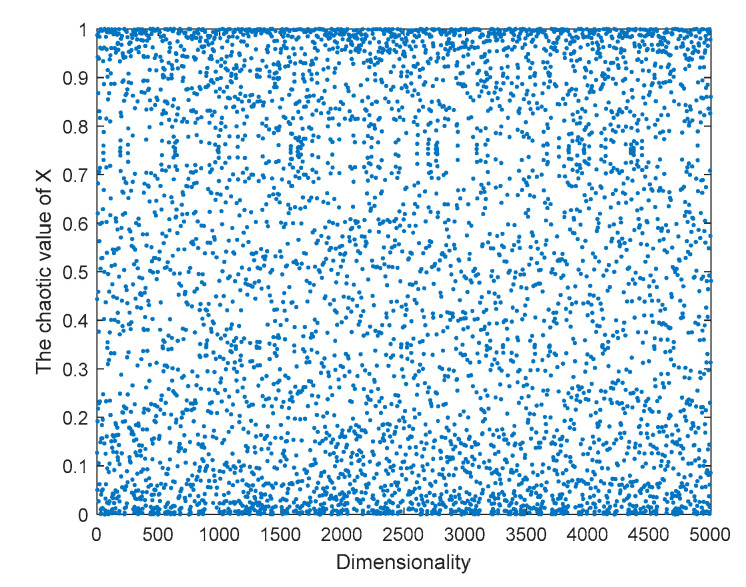
Logistic chaos value distribution.

**Figure 3 sensors-23-03988-f003:**
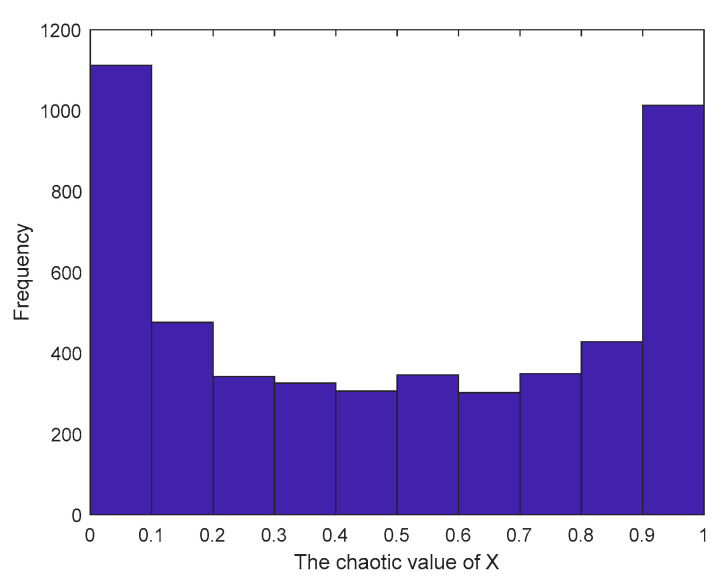
Logistic chaos value histogram.

**Figure 4 sensors-23-03988-f004:**
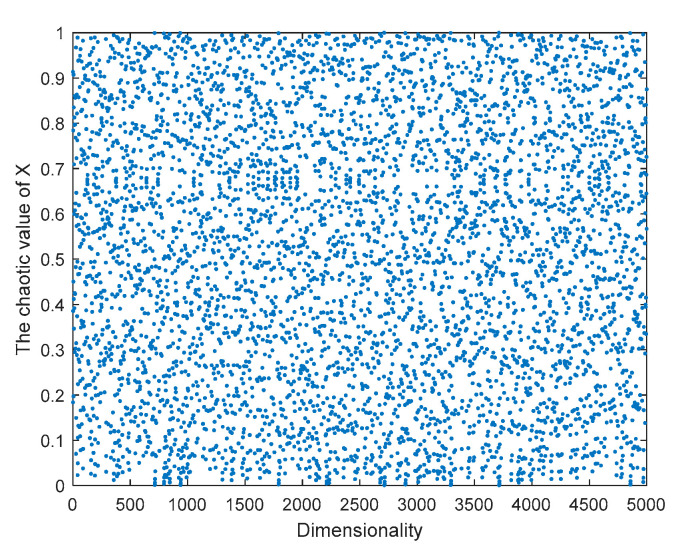
Distribution of improved logistic chaos values.

**Figure 5 sensors-23-03988-f005:**
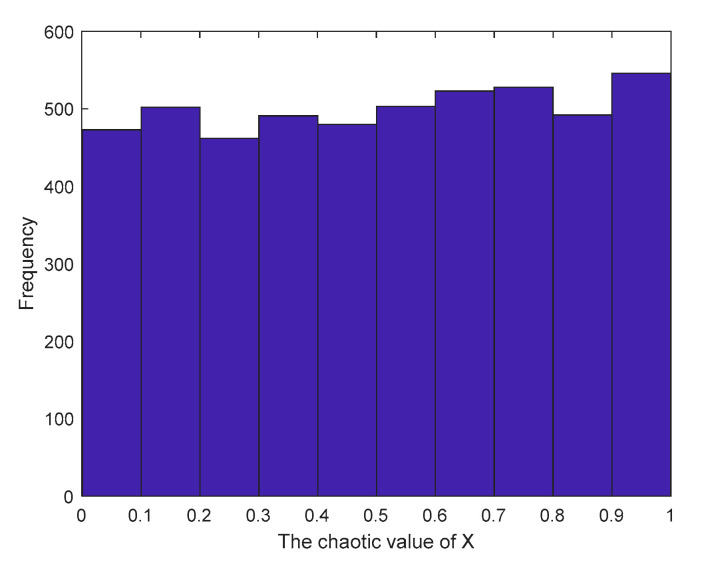
Improved logistic chaos value histogram.

**Figure 6 sensors-23-03988-f006:**
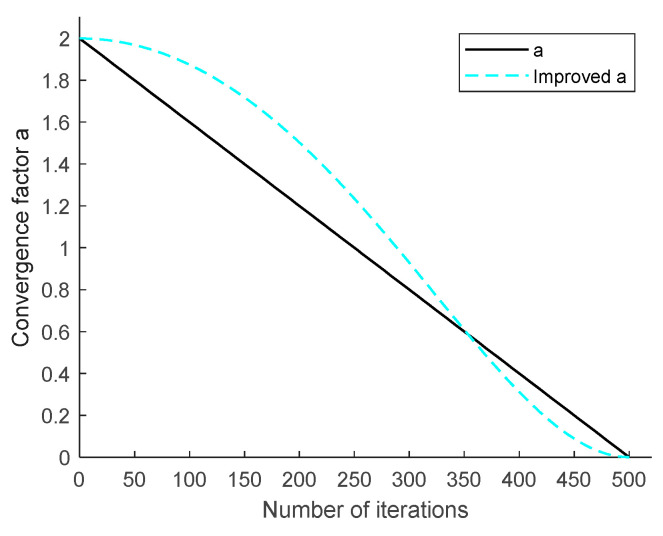
Graph of the variation of a with the number of iterations.

**Figure 7 sensors-23-03988-f007:**
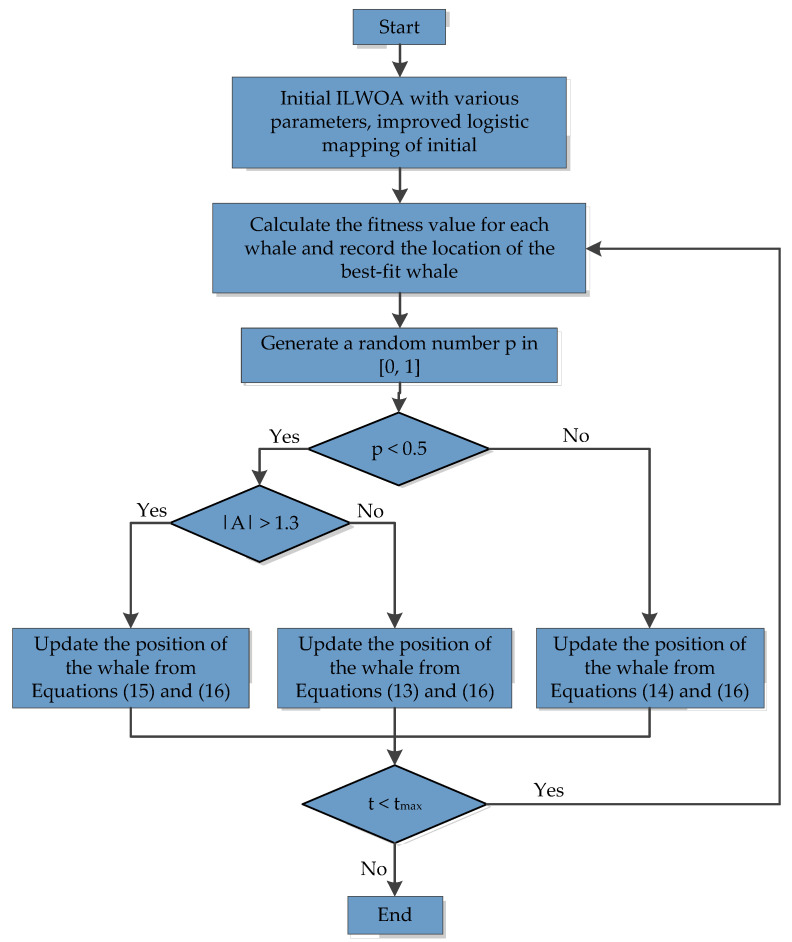
Flowchart of ILWOA.

**Figure 8 sensors-23-03988-f008:**
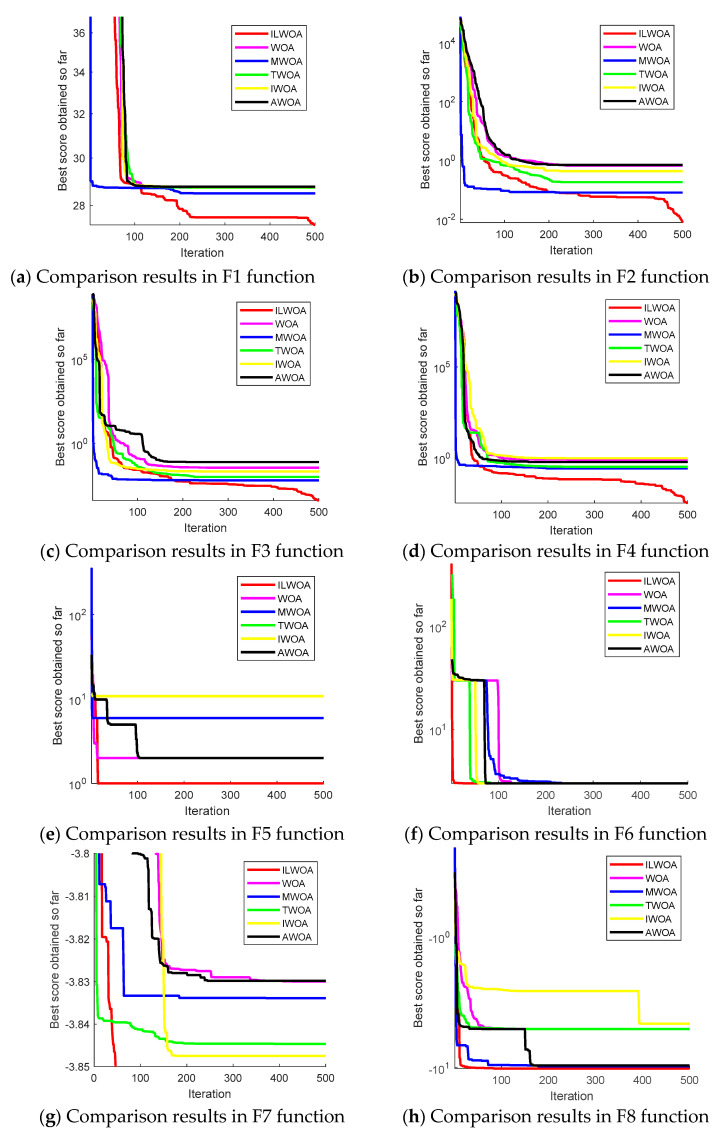
Curve chart based on eight test functions.

**Figure 9 sensors-23-03988-f009:**
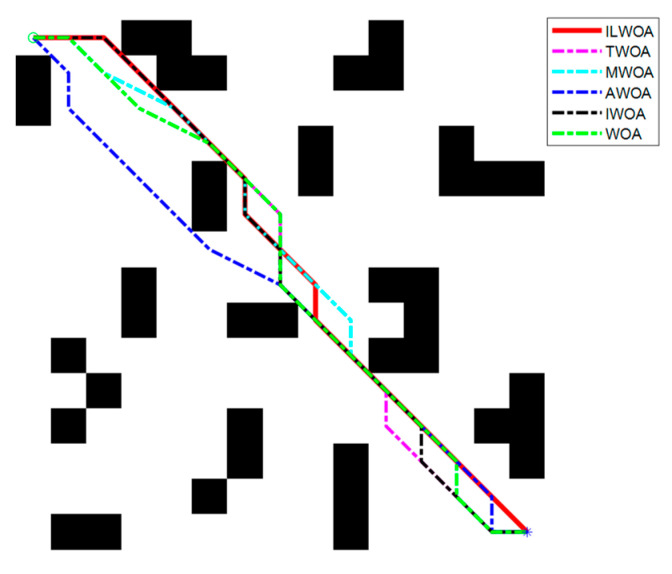
Robot path planning in Environment 1 scenarios.

**Figure 10 sensors-23-03988-f010:**
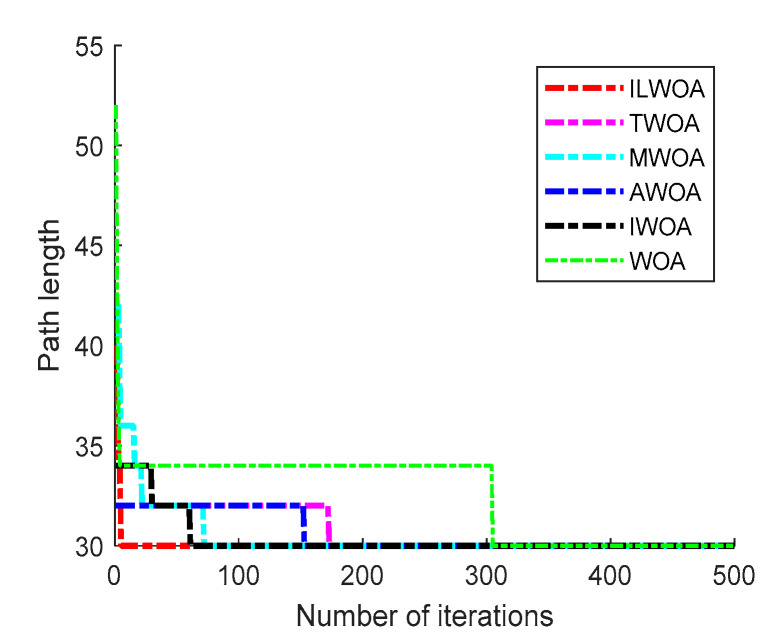
Optimal iteration curve.

**Figure 11 sensors-23-03988-f011:**
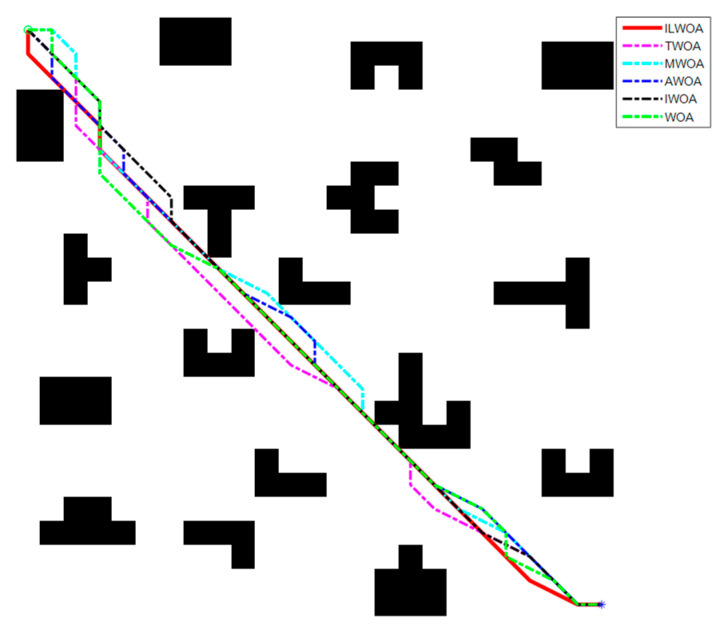
Robot path planning in Environment 2 scenarios.

**Figure 12 sensors-23-03988-f012:**
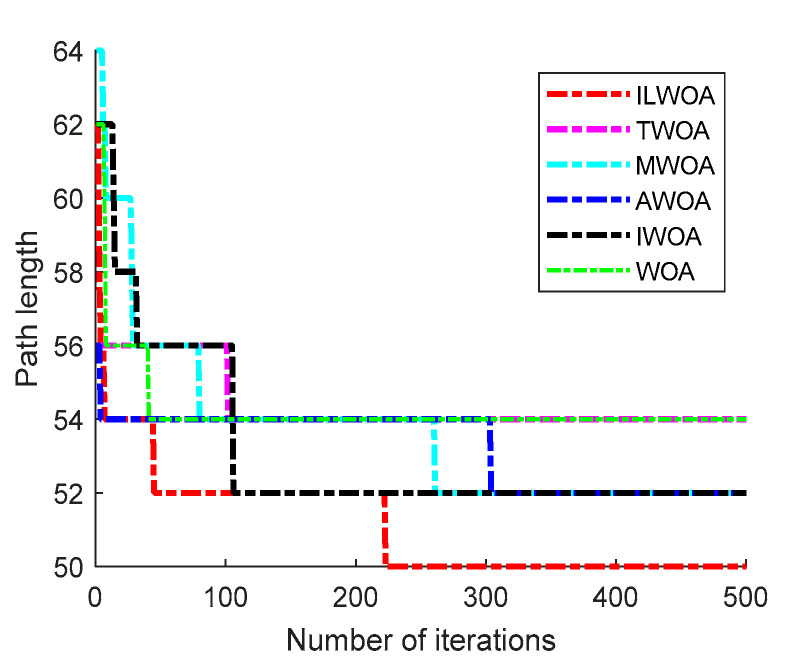
Optimal iteration curve.

**Figure 13 sensors-23-03988-f013:**
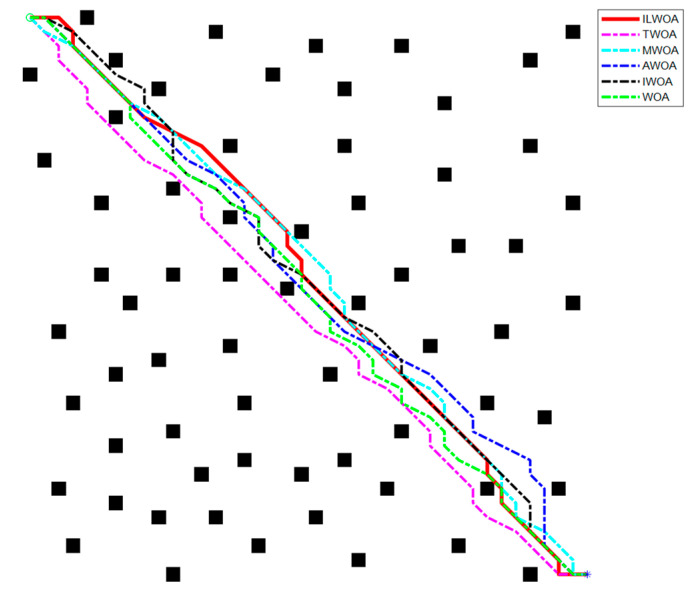
Robot path planning in Environment 2 scenario.

**Figure 14 sensors-23-03988-f014:**
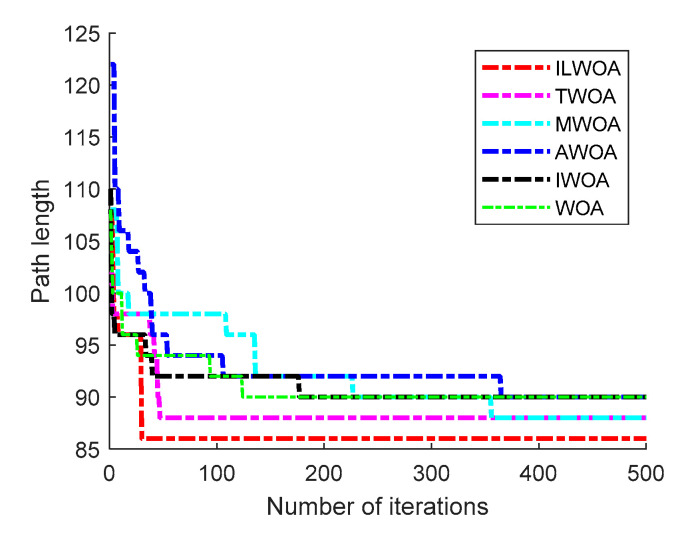
Optimal iteration curve.

**Table 1 sensors-23-03988-t001:** Algorithm parameter settings.

Parameter Name	Parameter Value
Population size N	30
Search Dimension dim	30
Maximum number of iterations $t_{max}$	500
Nonlinear factor minimum $a_{min}$	0
Nonlinear factor maximum $a_{max}$	2
Balancing parameter A in ILWOA algorithm	1.3
Remaining parameters of WOA algorithm	Referring to the literature [24]
Remaining parameters of MWOA algorithm	Referring to the literature [25]
Remaining parameters of TWOA algorithm	Referring to the literature [26]
Remaining parameters of IWOA algorithm	Referring to the literature [27]
Remaining parameters of AWOA algorithm	Referring to the literature [28]

**Table 2 sensors-23-03988-t002:** Standard test functions.

Standard Functions	Dimensionality	Search Space	Minimum Value
$F_{1}=\sum_{i=1}^{n-1}\left[{100\left(x_{i+1}-x_{i}^{2}\right)}^{2}+{\left(x_{i}-1\right)}^{2}\right]$	30	[−30, 30]	0
$F_{2}=\sum_{i=1}^{30}{\left(|x_{i}+0.5|\right)}^{2}$	30	[−100, 100]	0
$F_{3}=\frac{\pi }{30}\left\{10{sin}^{2}(\pi y_{1})+\sum_{i=1}^{29}{\left(y_{i}-1\right)}^{2}∙[1+10{sin}^{2}(\pi y_{i+1})]+{\left(y_{n}-1\right)}^{2}\right\}+\sum_{i=1}^{30}u\left(x_{i},10,100,4\right)$	30	[−50, 50]	0
$F_{4}=0.1\left\{{sin}^{2}(\pi {3x}_{1})+\sum_{i=1}^{29}{\left(x_{i}-1\right)}^{2}∙[1+{sin}^{2}(3\pi x_{i+1})]+{\left(x_{n}-1\right)}^{2}[1+{sin}^{2}(2\pi x_{30})]\right\}+\sum_{i=1}^{30}u(x_{i},5,100,4)$	30	[−50, 50]	0
$F_{5}={\left[\frac{1}{500}+\sum_{j=1}^{25}\frac{1}{j+\sum_{i=1}^{2}{\left(x_{i}-a_{ij}\right)}^{6}}\right]}^{-1}$	30	[−65.536, 65.536]	1
$F_{6}=\left[1+{\left(x_{1}+x_{2}+1\right)}^{2}\left(19-{14x}_{1}+{3x}_{1}^{2}-{14x}_{2}+{6x}_{1}x_{2}+{3x}_{2}^{2}\right)\right]\times \left[30+{\left({2x}_{1}-{3x}_{2}\right)}^{2}\left(18-{32x}_{1}+{12x}_{1}^{2}+{48x}_{2}-{36x}_{1}x_{2}+{27x}_{2}^{2}\right)\right]$	2	[−2, 2]	3
$F_{7}=-\sum_{i=1}^{4}c_{i}exp\left[-\sum_{j=1}^{4}{a_{ij}(x_{j}-p_{ij})}^{2}\right]$	4	[0, 1]	−3.86
$F_{8}=\sum_{i=1}^{5}{\left[(x-a_{i}){\left(x-a_{i}\right)}^{T}+c_{i}\right]}^{-1}$	4	[0, 10]	−10

**Table 3 sensors-23-03988-t003:** Experimental results of standard test functions.

Function Name	Statistical Results	ILWOA	WOA	MWOA	TWOA	IWOA	AWOA
F1	Average value	2.75 × 10^1^	2.79 × 10^1^	2.80 × 10^1^	2.80 × 10^1^	2.79 × 10^1^	2.82 × 10^1^
	Standard deviation	2.82 × 10^−1^	4.51 × 10^−1^	2.09 × 10^−1^	4.09 × 10^−1^	5.28 × 10^−1^	3.90 × 10^−1^
F2	Average value	1.96 × 10^−2^	4.53 × 10^−1^	1.96 × 10^−1^	4.25 × 10^−1^	5.87 × 10^−1^	6.19 × 10^−1^
	Standard deviation	8.30 × 10^−3^	2.66 × 10^−1^	7.42 × 10^−3^	2.46 × 10^−1^	3.38 × 10^−1^	3.40 × 10^−1^
F3	Average value	8.92 × 10^−4^	2.95 × 10^−2^	1.00 × 10^−2^	1.68 × 10^−2^	2.85 × 10^−2^	3.46 × 10^−2^
	Standard deviation	4.44 × 10^−4^	3.04 × 10^−2^	4.81 × 10^−3^	8.98 × 10^−3^	1.93 × 10^−2^	2.71 × 10^−2^
F4	Average value	3.31 × 10^−2^	4.90 × 10^−1^	1.47 × 10^−1^	5.22 × 10^−1^	5.71 × 10^−1^	5.48 × 10^−1^
	Standard deviation	1.91 × 10^−2^	1.93 × 10^−1^	8.41 × 10^−2^	3.19 × 10^−1^	2.33 × 10^−1^	2.87 × 10^−1^
F5	Average value	9.98 × 10^−1^	3.23	1.29	4.32	3.68	2.31
	Standard deviation	2.45 × 10^−11^	3.29	9.75 × 10^−1^	4.35	3.77	2.66
F6	Average value	3.00	3.00	3.03	3.00	3.90	3.00
	Standard deviation	4.51 × 10^−7^	1.86 × 10^−4^	8.94 × 10^−2^	2.58 × 10^−4^	4.95	7.18 × 10^−4^
F7	Average value	−3.86	−3.85	−3.84	−3.85	−3.86	−3.85
	Standard deviation	1.96 × 10^−5^	1.71 × 10^−2^	1.74 × 10^−2^	2.20 × 10^−2^	7.65 × 10^−3^	1.65 × 10^−2^
F8	Average value	−1.02 × 10^1^	−8.69	−1.01 × 10^1^	−8.18	−7.30	−9.04
	Standard deviation	7.75 × 10^−4^	2.45	1.17 × 10^−1^	2.63	2.91	2.27

**Table 4 sensors-23-03988-t004:** Experimental results of Environment 1 scenarios.

Evaluation Indicators	ILWOA	WOA	MWOA	TWOA	IWOA	AWOA
Average length of path	30.667	32.210	31.793	31.517	31.862	32.140
Path standard deviation	0.956	2.024	1.792	1.153	1.302	1.767
Average number of iterations	36	66	57	61	63	64

**Table 5 sensors-23-03988-t005:** Experimental results of Environment 2 scenarios.

Evaluation Indicators	ILWOA	WOA	MWOA	TWOA	IWOA	AWOA
Average length of path	53.467	58.533	57.530	57.400	56.867	56.670
Path standard deviation	1.479	4.725	2.956	2.851	3.848	3.252
Average number of iterations	73	93	83	79	82	86

**Table 6 sensors-23-03988-t006:** Experimental results of Environment 3 scenarios.

Evaluation Indicators	ILWOA	WOA	MWOA	TWOA	IWOA	AWOA
Average length of path	90.667	99.667	96.600	95.533	97.600	96.733
Path standard deviation	2.746	17.301	6.35	5.888	6.61	7.956
Average number of iterations	98	113	103	108	103	100

## Data Availability

Not applicable.

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
