# Peer review of "Indoor Robot Path Planning Using an Improved Whale Optimization Algorithm"

_sensors, 2023, doi:10.3390/s23083988_

Round 1
Reviewer 1 Report
The authors propose a modified whale optimization algorithm for the problem of indoor robot rount planning. The algotith description is detailed using formulas, flowchart and textual explanations. The results of testing the algorithm on benchmarks and comparison with analogues are given. The article is well structured, meets the requirements for Sensors.
Comments:
1. The introduction contains a detailed description of the whale algorithm and only 12 lines of the description of the indoor robot path planning, in which there are only 4 references. Taking into account the subjects of the Special Issue, it is necessary to significantly expand the part devoted to the review of indoor robot path planning.
2. Both lines in figure 6 have the same color.
3. Subheadings are not highlighted, for example, line 231 "Standard test function experiment".
4. The contribution of this research and a brief description of the next sections should be added at the end of the introduction.
5. There is no mathematical formulation of the indoor robot path planning problem. It is not clear how exactly the algorithm is used to solve it.
Author Response
Thank you for reviewing my manuscript and I have revised it to address the issues you raised. Here is the answer to your fifth question.The algorithm is applied to path planning, which is actually a test function solution. The test function solved by the algorithm is transformed into a map, and the process of solving is the process of finding the target point in the map, and the minimum value solved by the algorithm in the test function is the shortest feasible path found in the map.

Reviewer 2 Report
-It would be more accurate to give the definition about the whale optimization algorithm given in the introduction section under a separate heading and to mention about the general features of the whale optimization algorithm in the introduction section for comparison of pervious investigations.
-The introduction section should be outlined for broad description of latest papers regarding to other types whale optimization algorithm. In this context, it would be better if the differences between other studies were given in the form of a table. Table should be indicated the performance of the proposed algorithm performance rather than other types of algorithms.
- Figure 6 should be indicated with different colors for good understanding the differences.
- Picture quality of the image is very weak. Image quality should be improved as shown on Figure 7.
Author Response
Thank you for reviewing my manuscript and I have revised it to address the issues you raised.
Reviewer 3 Report
page2 - Caption for Figure 1 is on other page;
Page3 - row 91 equation number (1) must be on the right of the equation
row109 ith - please explain this term. It is nou t, t+1 ...
Page 6 row 211 Cauchy appears different
page 7 row 223 CUP is CPU
page 8 row 231 ''Standard test function experiment'' is a tile.
Figure 8 - I think it is too big for a single image (if it didn't fit o a page)
I also suggest to find a way to show a link between functions from table 2,3 and figure 8. Also explain the idea of best score.
Page 10 row 268 - a few iterations with fewer iterations ... It is not clear.
Page 10 row 254 Path planing sim.exp. is a title.
Please detail a little the link between the paper before row 254, and the last part.
Row 296 - please rearrange
Author Response

(The authors gave the same response as above.)

Round 2
Reviewer 1 Report
The authors have significantly improved the paper.
The mathematical description of Indoor robot path planning should be added.
Nevertheless the paper can be accepted
Author Response
Thank you very much for your comment. We have revised it and marked it blue.
